# Appropriate Application Methods for Salicylic Acid and Plant Nutrients Combinations to Promote Morpho-Physiological Traits, Production, and Water Use Efficiency of Wheat under Normal and Deficit Irrigation in an Arid Climate

**DOI:** 10.3390/plants12061368

**Published:** 2023-03-19

**Authors:** Majed Alotaibi, Salah El-Hendawy, Nabil Mohammed, Bazel Alsamin, Yahya Refay

**Affiliations:** Department of Plant Production, College of Food and Agriculture Sciences, King Saud University, Riyadh 11451, Saudi Arabia

**Keywords:** field conditions, foliar application, production efficiency, relative water content, chlorophyll pigments, soil application, *Triticum aestivum*, water-scarce countries

## Abstract

Freshwater shortage and inadequate nutrient management are the two major challenges for sustainable wheat production in arid agro-ecosystems. Relatively little is known about the positive roles of the application methods for the combination of salicylic acid (SA) and plant nutrients in sustaining wheat production under arid climatic conditions. A two-year field study was undertaken to assess the impact of seven treatments for the integrated application of SA, macronutrients, and micronutrients on the morpho-physiological traits, yield, and irrigation water use efficiency (IWUE) of wheat subjected to full (FL) and limited (LM) irrigation regimes. The results showed that the LM regime caused a significant reduction in different plant growth traits, relative water content, chlorophyll pigments, yield components, and yield, while a significant increase was observed in IWUE. The sole application of SA or co-application with micronutrients through soil did not significantly affect the studied traits under the FL regime, while they achieved some improvement over untreated plants under the LM regime. Based on the different multivariate analyses, the soil and foliar applications for the combinations of SA and micronutrients, as well as a foliar application for the combinations of SA, macronutrients, and micronutrients were identified as an efficient option for mitigating the negative impacts of water deficit stress and enhancing the growth and production of wheat under normal conditions. In conclusion, the results obtained herein indicated that the co-application of SA and macro- and micronutrients is an effective option to greatly enhance and improve the growth and production of wheat crops in water-scarce countries of arid regions, such as Saudi Arabia, while an appropriate application method for this combination was required for positive effects.

## 1. Introduction

Water scarcity is one of the significant environmental issues that restrict food crop production and affect all factors related to ensuring global food security. Unfortunately, several studies have predicted that water scarcity will increase due to continued human activity and increased emissions leading to global warming and climate change, particularly in arid and semiarid regions [1,2]. It is predicted that water scarcity will affect more than 50% of global croplands by 2050. Therefore, food crop production is still far below the expected quantities to meet the increase in food demand. 

Bread wheat (*Triticum aestivum* L.) is one of the most important cereal crops produced and consumed worldwide. It provides approximately 20% of the protein and daily calories for 4.5 billion people around the world. It is grown on a total area of approximately 219 million ha, with a total production of over 760 million tons in 2020 [3]. Demand for wheat is estimated to increase by 60% by 2050, posing a serious concern for meeting that demand [4]. Additionally, global warming will cause severe drought in 60% of the world’s wheat-growing regions by the end of this century. Recently, drought affects 15% of wheat productivity [5]. In addition, water deficit is considered a predominant factor, causing a significant yield loss of the wheat crop due to its negative effects on several morpho-physiological traits during their critical growth stages [6]. It was reported that the exposure of wheat plants to deficit water stress at the tillering stage results in a significant reduction in plant height, tiller number, spiked tillers, leaf area, photosynthesis activity, chlorophyll content, several metabolic processes, and relative water content (RWC), which eventually leads to a sharp decrease in plant biomass [7,8]. The flowering and grain-filling stages are considered the most sensitive stages to water deficit stress in wheat. If water stress occurs at both stages, this can lead to a significant decrease in spike weight, number and weight of grains per spike, and thousand-grain weight, which eventually may cause yield losses by more than 50% [9,10]. As a consequence, supplementary strategies are urgently needed to lessen the negative impacts of water deficit stress on the growth and production of wheat crops in water-scarce countries of arid regions.

Several strategies, which are based on chemical, physical, and biological techniques, have been suggested to mitigate the negative impacts of water deficit stress on the growth and production of plants. Among the strategies associated with the introduction of tolerance toward water deficit stress is the exogenous application of macronutrients and micronutrients, as well as compatible solute compounds, such as salicylic acid. In general, previous studies highlighted that the exogenous application of these materials is considered a cost-effective and efficient option to alleviate the harmful effects of abiotic stress on the growth and production of crop plants [7,8,11].

The application of plant nutrients contributes to manipulating several environmental stresses when applied in a proper manner. The availability of suboptimal macronutrients and micronutrients retards enzymatic activities and the normal functions of different physiological and biochemical processes. Now, there are several macronutrients and micronutrients considered to be important for the optimum growth, development, and productivity of plants because they are activators and/or cofactors of several physiological and metabolic processes in plants. They play several roles in plants, including the synthesis of antioxidants and DNA, production of amino acids for protein synthesis, translocation of photosynthates from source (leaves) to sink (grains), regulation of stomatal movement and energy production, stimulation of adventitious roots and cell division, maintenance of membrane structures, cell turgidity, osmotic adjustment, root cell membrane integrity, and chloroplast structure and functions, avoidance of the osmotic stress induced by environmental stresses, and act as an activator and cofactor of hundreds of metalloenzymes in plants, which eventually help plants in alleviating various abiotic stresses [12,13,14,15,16,17,18,19,20]. Therefore, the foliar and/or soil application of these nutrients could be used for different purposes, including alleviating the adverse effects of water deficit stress, as well as achieving higher productivity, quality, and profitability under both normal and stress conditions. For instance, Karim and Rahman [21] reported that the foliar application of micronutrients (Zn, B, and Mn) along with the soil application of macronutrients (NPK) significantly reduced the negative impacts of drought stress that often occur during the late stages of cereal production in Asia’s least-developed countries. They also found that the soil application of micronutrients in the early stage, coupled with foliar application in the late stage, is a promising option to mitigate drought stress on cereal production. Hussain et al. [22] found that the foliar application of N and K promoted plants of the sunflower crop to water deficit stress by maintaining cell turgor, improving the accumulation of the osmoprotectants, increasing the net photosynthesis rate and stomatal conductance, and decreasing the production of reactive oxygen species (ROS), which ultimately resulted in a significant increase in yields. As reported by Barłóg et al. [23], the foliar or soil application of Zn significantly improves the growth, production, and quality of sugar beets under normal growth conditions by balancing translocation and nutrient uptake. Dass et al. [24] also reported that applying a small amount of macronutrients and micronutrients through foliar sprays, along with a recommended dose of fertilizers, significantly improved the growth, yield, and WUE of soybeans under semiarid conditions. In contrast, the application of macronutrients through foliar sprays, particularly N, can lead to a decrease in the yield of crops due to severe leaf scorching, which often occurred following the foliar application of N [25,26]. In addition, the foliar application of different plant nutrients is affected by several factors, including air temperature, the amount of rain, light quality and intensity, nutrient mobility, and leaf characteristics [25]. On the other hand, excessive bicarbonate and high pH in arid soil conditions affect the availability of several nutrients to plants. Therefore, despite the limitations of the foliar application method, under certain circumstances, this method is the most effective way to correct the nutritional disorder within a relatively short time, and they are therefore especially efficient as preventive and, in some cases, curative treatments [27]. Not much information is available about the appropriate methods for the application of macronutrients and micronutrients to enhance the growth, production, and WUE of wheat under arid conditions. 

When plants are exposed to water deficit stress, several physiological processes are disrupted by increasing the production of ROS, which leads to oxidative stress. This oxidative stress results in inhibiting chlorophyll biosynthesis, ion homeostasis, hormonal balance, photosynthetic activity, DNA damage, and imbalances in nutrient and water relations [28,29,30]. Therefore, plants can deal with water deficit stress through different physiological and biochemical processes. Among the latter, some plants can accumulate and synthesize different compatible solutes or osmolytes to modulate the negative impacts of water deficit stress on their growth and production by enhancing antioxidant activities and scavenging ROS. Among these compatible solutes, salicylic acid (SA) is one of the essential phytohormones that can regulate multiple physiological and biochemical processes under different environmental stress conditions. It can play a vital role in the defense mechanism against water deficit stress. Previous research has pointed out that the exogenous application of SA enhances the morpho-physiological traits and yield of different field crops under different environmental stress conditions, including drought stress. It has been also reported that SA improves plant tolerance to drought stress by reducing ROS production, maintaining osmotic adjustment, regulating stomatal movement, improving the photosynthetic capacity, and stimulating nutrient uptake [31,32,33,34,35].

Previous studies have also reported that the integration of SA with plant nutrients can achieve good results in the response of plants to abiotic stresses. For instance, Yavas and Unay [36] found that the foliar application of SA and Zn together significantly reduced the negative impacts of drought stress on several plant traits of wheat, such as chlorophyll content, RWC, plant height, spike length, number of grains per spike, and grain weight. In a pot experiment, Noori et al. [37] found that the foliar application of SA combined with soil application of K at high levels (200 kg K ha^−1^) increased the production of wheat under drought stress conditions. Munsif et al. [38] also found that foliar application of SA in combination with the soil application of K had substantially reduced the negative impacts of mild (60% of field capacity) and severe (30% of field capacity) drought stresses on plant water status, growth, production, and the bio-physiological characteristics of wheat. 

However, the effect of the combined application of SA and plant nutrients through the soil on wheat performance under arid conditions has yet to be studied. This study hypothesized that the application methods of combinations between SA and plant nutrients may play an important role in enhancing the performance of wheat under normal and stress conditions. Therefore, the overall goal of this study was to identify the appropriate application methods for the combinations between SA, macronutrients and micronutrients to achieve maximum growth, yield, and IWUE for wheat under sufficient and limited irrigation water supply conditions in an arid agro-ecosystem. 

## 2. Results

### 2.1. Growth Parameters 

The analysis of variance showed that the F-values, due to irrigation regimes (IR) and different combination treatments between SA, macronutrients and micronutrients (T), were significant at *p* ≤ 0.05 or 0.01 for all growth parameters in each growing season and the combined two seasons, except for plant height (PH) in the second season and the combined two seasons and tiller number (TN) in the two seasons (Table 1). All growth parameters in both growing seasons and the combined two seasons, except PH and TN, were significantly affected at *p* ≤ 0.01 by the interaction IR × T. Seasons (S) had an insignificant effect on all growth parameters, except shoot dry weight (SDW). The interaction IR × S, T × S, and IR × T × S showed insignificant effects on all growth parameters at *p* ≤ 0.05 or 0.01, except TN, green leaf number (GLN), leaf area index (LAI), and shoot fresh weight (SFW), which were significantly affected by IR × S interaction (Table 1). 

Averaged over the two seasons, the limited irrigation regime (LM) hampered PH, TN, GLN, LAI, SFW, and SDW by 12.2%, 21.5%, 43.6%, 47.9%, 35.7%, and 32.1%, respectively, when compared with the full irrigation regime (FL) (Table 2). However, the exogenous applications of different T substantially improved all the parameters mentioned above as compared to the non-treated control treatment (T1). In general, T4 (the soil and foliar application of SA and micronutrients) exhibited the highest values for all growth parameters followed by T7 (foliar application of SA, macronutrients, and micronutrients), while the T1 exhibited the lowest ones (Table 2). Over the two seasons, the T2 (soil application of SA), T3 (soil application of SA and micronutrients), T4, T5 (soil application of SA and foliar application of macronutrients and micronutrients), T6 (foliar application of SA), and T7 increased GLN by 4.3–41.3%, LAI by 10.5–53.3%, SFW by 5.1–26.0%, and SDW by 6.3–27.1%, as compared to T1 (Table 2). 

The responses of growth parameters to different T under each IR regime over both growing seasons are shown in Figure 1. The different T (T2−T7) improved the LAI, GLN, SFW, and SDW under FL by 1.9–41.1%, 1.6–43.2%, 2.0–26.3, and 2.6–23.8%, respectively, in comparison to T1. The maximum increase in the parameters mentioned above was observed with T7 and T4, which increased by 22.5–40.7% in comparison to their corresponding T1. However, the T2 and T3 did not improve the aforementioned growth parameters under the FL regime, and the values of these parameters for both treatments were statistically on par with the T1 (Figure 1). Contrarily, all T improved the parameters mentioned above under the LM regime, and T4 showed a maximum increase in these parameters by 36.8–74.2%, as compared with the T1 (Figure 1).

### 2.2. Physiological Parameters 

The IR regime, T, and S had a significant main effect on all physiological parameters, namely relative water content (RWC), chlorophyll a (Chla), chlorophyll b (Chlb), and total chlorophyll content (Chlt) in each growing season as well as the combined two seasons (Table 3). Additionally, the four physiological parameters were significantly affected by the interaction IR × T, except Chlb in the first season, IR × S, and T × S, except RWC. The interaction effect of IR × T × S had a significant effect on Chla at the 5% probability level and a non-significant effect on other physiological parameters (Table 3). 

Irrespective of T, averaged over both seasons, the LM regime significantly reduced RWC, Cha, Chb, and Cht by 22.4%, 39.8%, 40.8%, and 41.1%, respectively, as compared to the FL regime (Table 4). Regarding the effect of different T, the T4 showed the highest values for all physiological parameters, followed by T7. Averaged over both seasons, the T4 and T7 enhanced 12.0% and 7.4% more RWC, 32.7% and 30.6% more Chla, 33.3% and 29.0% more Chlb, and 32.7% and 29.9% more Chlt as compared to the T1, respectively (Table 4). Similarly, the other treatments (T2, T3, T5, and T6) were accompanied by increases in RWC by 5.9–8.7%, Chla by 15.9–24.4%, Chlb by 17.0–21.9%, and Chlt by 16.0–23.2% compared to the T1 (Table 4).

The responses of physiological parameters to different T under each irrigation regime are presented in Figure 2. Under the FL regime, plants treated with foliar application (T6 and T7 treatments) or combined foliar and soil applications (T4 and T5 treatments) showed significant superiority in RWC over those plants treated with soil application only (T2 and T3 treatments) or untreated plants (T1). However, under the LM regime, the T2–T7 treatments significantly outperformed the T1 in RWC by 7.9–25.4% (Figure 2). Regarding chlorophyll pigments under the FL regime, the highest values for these parameters were achieved with T7 and T4, followed by T3, T5, and T6, while the T2 were statistically on par with the T1. However, under the LM regime, the T2–T7 exhibited higher chlorophyll pigments than those of the T1, with the treatments of the foliar application or a combination of soil and foliar applications were better than those treatments of sole soil application in enhancing the chlorophyll pigments (Figure 2).

### 2.3. Yield and Yield Attributes Parameters 

The F-values of the ANOVA analysis showed that all yield components and irrigation water use efficiency (IWUE) were significantly affected by the IR regime, T, and their interaction, with the exception of spike length (SL), in which the T and IR × T interaction had non-significant effects on this parameter (Table 5). The S showed a significant effect on only the grain weight per spike (GWS), number of grains per spike (NGPS), and harvest index (HI). The IR × S interaction showed non-significant effects on any yield parameters and IWUE, and the T × S and IR × T × S interaction showed significant effects on only HI (Table 5). 

Averaged over both seasons, the LM regime significantly reduced SL, GWS, GNPS, thousand grain weight (TGW), grain yield per ha (GY), biological yield per ha (BY), and HI by 16.2%, 38.5%, 26.5%, 16.5%, 41.8%, 29.1%, and 18.1%, respectively, while it increased irrigation water use efficiency (IWUE) by 14.1%, when compared with the FL regime (Table 6). Regardless of the IR, the T2−T7 treatments considerably enhanced GWS, GNPS, TGW, GY, BY, HI, and IWUE over the two seasons by 6.4–30.5%, 1.4–14.0%, 6.3–17.1%, 12.7–39.0%, 9.0–26.8%, 5.0–11.8%, and 16.5–45.3%, respectively, as compared with the T1 (Table 6). The maximum increases in yield components, yield, and IWUE were recorded with T4, followed by T7. Averaged over two seasons, the T4 and T7 increased the different parameters of yield and IWUE by 17.1–45.3% and 7.2–34.5%, respectively, in comparison to the T1, with the Duncan’s test in decreasing order of these parameters were T4 > T7 > T6 > T5 > T3 > T2 > T1 (Table 6).

The responses of yield components, yield, and IWUE to the exogenous applications of combination treatments under each irrigation regime are shown in Figure 3. Under the FL regime, the maximum values for yield parameters and IWUE were obtained with the treatments of foliar application (T6 and T7) and a combination of foliar and soil application (T4 and T5), with the T4 and T7 always exhibiting the highest values for these parameters. The treatments of soil application only (T2 and T3) are less efficient than the other combination treatments in improving the yield parameters and IWUE, and the values of these parameters for both treatments were statistically on par with the T1 under the FL regime (Figure 3). However, under the LM regime, all combination treatments improved yield parameters and IWUE compared to the T1. Compared with the T1, the increases in GWS, NGPS, TGW, GY, BY, HI, and IWUE due to treated plants with different combination treatments (T2–T7) was 4.2–2.7%, 13.9–30.5%, 28.1–64.6%, 16.1–35.5%, 10.6–22.1%, and 28.1–64.6%, respectively (Figure 3). Overall, the treatment of a combination of soil and foliar applications (T4) recorded the highest values for all yield parameters and IWUE and statistically outperformed the other treatments, except for TGW and BY (Figure 3).

### 2.4. Pearson’s Correlation Coefficient among Different Parameters under the Full and Limited Irrigation Regimes

In Table 7, which shows the Pearson’s correlations values between growth, physiological, yield components, yield, and IWUE parameters under each IR regime, all the studied parameters showed a significant positive correlation (r = 0.77–0.99) with each other, except PH and SL under both irrigation regimes and TN under the FL regime, which displayed a weak and occasionally even a negative correlation with all parameters. Furthermore, there were positive and significant correlations of GY and IWUE with all studied parameters (r = 0.78–0.99) except PH, TN, and SL under the FL regime. Similarly, GY and IWUE were strongly positively correlated with all studied parameters (r = 0.77–0.99) except SL under the LM regime. The RWC and Chlt showed significant positive correlation with almost all growth and yield parameters under both irrigation regimes (Table 7).

### 2.5. Principal Component Analysis and Heatmap Analysis for the Responses of Plant parameters to Different Combination Treatments

The associations between the 16 studied parameters and 7 combination treatments under each irrigation regime were evaluated via a principal component analysis (PCA) scatter plot (Figure 4) and heatmap clustering (Figure 5). The first two principal components (PC1 and PC2) explained 85.94% and 92.10 of the total variations derived from the 16 parameters under the FL and LM regimes, respectively (Figure 4). The PC1 explained 76.59% and 85.90% of the total variance, while the PC2 explained 9.35% and 6.19% of the total variability between all parameters under the FL and LM regimes, respectively. Both PCs grouped all studied parameters together in one group in a positive direction of PC1 under both irrigation regimes, with the angle between the vector of these parameters being acute, which indicates a positive association between these parameters (Figure 4). Additionally, all parameters were strongly positively associated with T4 (first quarter), T7 and T6 (second quarter) under the FL regime (Figure 4) and T4 (first quarter), T7, T6, and T5 (second quarter) under the LM regime (Figure 4). The combination treatments in the first and second quarters of PCA were leaned toward the highest PC1 and lowest or highest PC2. All studied parameters did not show any associations with T1, T2, and T3 under both irrigation regimes, in addition to T5 under the FL regime (Figure 4).

Figure 5 shows that the heatmap dendrogram distributed the different combination treatments into three and four groups under the FL and LM regimes, respectively. The T4 and T7 under the FL regime and T4 alone under the LM regime were clustered into one group and displayed the highest values for all studied parameters under FL, except the PH and TN, and the LM regimes (Figure 5). The T1 was clustered alone in one group under the LM regime, while it clustered with T2 and T3 (treatments of soil application only) in one group under the FL regime. The T5 and T6 were clustered together in one group under the FL regime, while they clustered with the T7 treatment in one group under the LM regime (Figure 5). 

## 3. Discussion

In this study, all studied parameters were significantly affected by irrigation regimes (Table 1, Table 3 and Table 5) and the LM regime considerably reduced the growth, physiological, and yield parameters compared with the FL regime (Table 2, Table 4 and Table 6). These reductions in the different studied parameters under the LM regime may be due to the exposure of plants to water deficit stress, resulting in substantial inhibition in several morphological, physiological, and biochemical attributes, including limiting RWC, photosynthetic rate, stomatal conductiance, activity of photosynthetic enzymes, chlorophyll biosynthesis, cell division, cell expansion, and biomass accumulation. Additionally, water deficit stress indirectly affects plant growth and production by hindering the access of nutrients to the plant and reducing their mobility and absorption even with their availability in the soil [39,40,41]. A shortage of nutrients, or at least some of them, leads to physiological and biochemical disturbances in the plant as a result of the role of these elements in important vital processes, such as cell osmotic relations and turgor-related processes, photosynthesis reactions, enzymatic activity, building nucleic acids, and plant reproduction [12,15,42]. Furthermore, water deficit stress induces oxidative damage that may affect plant water relations, leaf pigments, and cell membrane integrity as well as leading to the generation of ROS [43,44]. Therefore, previous studies have reported that the synthesis and accumulation of compatible solutes are considered one of the most common methods of maintaining plant water relations and removing excess levels of ROS under water deficit stress [31,32,33,34,35]. SA is one of the most promising of these compatible solutes in enhancing the growth and production of crops under water deficit stress by enhancing osmotic regulation, reserving water in plant cells, promising water potential gradients, enhancing the activities of antioxidant enzymes, concealing the ROS, and regulating the uptake and assimilation of many nutrients [34,35,45]. Therefore, we hypothesized in this study that the exogenous co-application of SA, macronutrients, and micronutrients through soil and/or foliar spraying methods could be considered as a cost-effective and easy strategy for not only mitigating the water deficit stress on wheat growth and production but also enhancing the performance of wheat crop under normal irrigation conditions in arid agro-ecosystems.

The current study showed that the combined application of SA, macronutrients, and micronutrients through soil and foliar spray methods improved the performance of wheat under the FL regime as well as having the ability to ameliorate the negative impacts of water deficit stress on morpho-physiological traits and the production of wheat under LM regimes (Table 2, Table 4 and Table 6). Several previous studies indicated the efficacy application of SA and/or nutrients, such as N, P, K, Zn, and Mn, to enhance the growth and yield of many crops such as wheat, rice, soybeans, mung beans, and maize under water deficit stress and reduce the adverse effects of water shortage on the production of these crops [33,46,47,48,49]. In this regard, El Sherbiny et al. [33] concluded that the foliar application of SA at 700 µM could be used to improve the growth, production, and WUE of rice as well as to mitigate the negative impacts of limited water irrigation. Razmi et al. [46] also found that spraying soybean plants exposed to water deficit stress with SA improved the parameters of LAI, RWC, and chlorophyll a and b. Similarly, Ahmad et al. [47] indicated that the exogenous application of SA in wheat enhanced the leaf water potential, osmotic potential, chlorophyll contents, photosynthetic rate, RWC, and antioxidant enzymatic activities under both normal and stress conditions. On the other hand, Waraich et al. [50] reported that application of some macronutrients and micronutrients, such as N, P, K, Zn, Si, and Mg, can reduce the negative impacts of drought stress on plants. There are several mechanisms by which these elements reduce the negative impacts of water deficit stress as well as enhance the performance of plant crops under normal conditions. For example, foliar-applied fertilizer of N enhanced the process of photosynthesis and regulated several metabolic processes in plants as well as enhancing the leaf Chl contents and the amount of photosynthetically active radiation intercepted by plants, which eventually leads to further improving plant growth and yield parameters [24]. K is involved in a wide range of plant processes, including photosynthesis, stomatal regulation, enzyme activity, osmoregulation and membrane stability, carbohydrate synthesis, and the transportation of assimilates from source to sink, which all are important to enhance the growth and production of crops under both normal and stress conditions [14,15,22]. P is essential for numerous cellular functions, including energy transfer, enzyme activation, membrane structure maintenance, biomolecule synthesis, and the creation of high-energy molecules (adenosine triphosphate). Moreover, it enhances root development, resulting in increased water uptake and enhancing the growth and productivity of crop plants in water-scarce environments [13,51,52,53]. Additionally, foliar-applied micronutrients, such as Zn and Mn, enhanced the growth and production of several field crops under both normal and water stress conditions because both ions are involved in a wide range of developmental and physiological processes, including enhancing auxin formation, cell expansion, stem elongation, source–sink relationship, membrane stability, hormone and chlorophyll synthesis, enzyme activation, and antioxidant defense [18,20,54,55]. Furthermore, the exogenous application of SA, which also is involved in several physiological, biochemical, and development processes in plants, also play an important role in regulating plant growth and development under both normal and stress conditions. It improves the growth and production of crops by stimulating the cell division, photosynthetic activity, and water potential gradients, regulating some physiological responses related to carbon uptake and/or fixation, increasing the creation of photosynthetic pigments, raising stomatal conductance, delaying the senescence of plant organs, and regulating the source-to-sink relationship [33,34,35].

In this study, the response of different morpho-physiological and yield parameters of wheat to the co-application of SA, macronutrients, and micronutrients varied according to the irrigation regimes (Figure 1, Figure 2 and Figure 3). Under the FL regime, the co-application of SA, macronutrients, and micronutrients through foliar application (T7) or foliar and soil applications (T4) were more efficient for enhancing the different studied parameters than those of sole application of SA (T1 and T6), sole application of SA through soil with the sole application of nutrients through foliar (T5), or the co-application of SA and micronutrients through soil application only (T3), while the T2 and T3 did not show any significant variation from T1 (Figure 1, Figure 2 and Figure 3). However, under the LM regime, the T4 achieved the highest values for all studied parameters, followed by T5, T6, and T7. Additionally, the T2 and T3 achieved higher values for the most studied parameters than the T1 under the LM regime (Figure 1, Figure 2 and Figure 3). These results indicate that the co-application of SA with micronutrients through both soil and foliar spray methods (T4) is a useful and simple approach to enhance the growth and production of wheat crops under both normal and water deficit stress conditions. This means that applying a small amount of SA and micronutrients through foliar sprays directly on the foliage, in addition to applying a low dose of these materials through soil application, are efficient approaches to improving the growth, physiological, production, and IWUE of the wheat crop when compared with using only one mode of application for these materials. As the roots are the first organ to experience water deficit stress, in addition to the leaves being the key organ in which the majority of biochemical and physiological processes occur, this may explain why the combination of soil and foliar applications methods for the combinations of SA and micronutrients seems to be an effective approach for enhancing the growth and production of wheat under normal and water deficit stress conditions. In addition, the foliar application of SA and micronutrients helps their entry directly into the leaves, which raises their concentration in them sufficiently. This helps protect various biochemical and physiological processes from the negative effects of water deficit stress, leading to rapid relief of physiological stress, and also promoting photosynthesis and regulating other metabolic processes in the plant. While the application of SA and micronutrients through soil significantly raises their concentration in the roots, which enhances root growth under both normal and stress conditions as well as improving the osmotic adjustment in roots under stress conditions. This enables the roots to extract water and nutrients from deeper soil layers and the presence of SA as compatible solutes in the roots and micronutrients, such as Zn and Mn, enhance water uptake and promising water potential gradients under water deficit stress conditions. These findings are in line with the previous study conducted by Karim and Rahman [21], who reported that the soil application of micronutrients (Zn, B, and Mn) at an early stage of cereal crops in combination with foliar application at a late stage is a promising approach to alleviating the negative impacts of drought stress, which often occur during the late stages of cereal production in Asia’s least developed countries. They also found that the foliar application of micronutrients (Zn, B, and Mn) along with the soil application of macronutrients (NPK) significantly enhanced cereal production under drought stress conditions. 

The results of this study also found that the co-application of SA, macronutrients,- and micronutrients through foliar sprays only (T7) is a more efficient strategy for enhancing the performance of wheat crops under the FL regime than under the LM regime, as shown with the heatmap clustering for different combination treatments under the FL and LM conditions (Figure 5). This finding indicates that the foliar application of macronutrients and micronutrients can be considered an essential strategy for achieving a higher productivity of wheat under normal conditions. This result can be attributed to the fact that the foliar application of nutrients can increase the efficiency of nutrient absorption and delay leaf senescence; then, photosynthesis can be enhanced and/or prolonged. Similarly, improved productivity, quality, and profitability in soybean crops were reported by Dass et al. [24] and Gheshlaghi et al. [56] using the foliar application of essential nutrients under normal conditions. However, the soil application of micronutrients has been noticed to be very effective in alleviating the adverse effects of water deficit stress on wheat performance under the LM regime. Similarly, Ma et al. [57] found that the soil application of Zn led to an increase in wheat grain yield under stress conditions as well as under adequate water supply. This increase was directly proportional to the intensity of the stress and was 28.2%, 22.6%, and 10.5% in severe drought, moderate drought, and adequate water supply, respectively. The same was reported by Dimkpa et al. [58], who found that the soil application of micronutrients was more effective in reducing the negative effects of water deficit stress on soybeans.

It is observed from this study that yield, yield components, and IWUE were significantly and strongly correlated with each other and with the different morpho-physiological parameters under both irrigation regimes (Table 7). This result indicates that the co-application of SA, macronutrients, and micronutrients positively affected several morpho-physiological and biochemical aspects of wheat plants under both water and non-water deficit conditions. This may be because the integration between SA and plant nutrients may play a defensive role in mitigating the negative impacts of water deficit stress on wheat performance under stress conditions while playing a synergistic role in enhancing wheat performance under normal conditions. Therefore, a statistically significant correlation was observed between most studied traits under the FL and LM regimes (Table 7). Similarly, Dass et al. [24] and El Sherbiny et al. [33] reported that strong and positive correlations were found between most studied parameters of soybean and rice crops under both water and non-water deficit conditions due to the exogenous application of SA, macronutrients, and micronutrients. This is because the SA and/or macronutrients, and micronutrients are involved in a wide range of developmental and physiological processes, not only under stress conditions but also under normal conditions. For instance, plants of many species treated with SA increase their concentrations of essential elements such as N, P, K, Ca, S, Fe, Mn, and Mg [59,60,61]. This mineral nutrient ultimately results in a significant enhancement in the chlorophyll a, b, and carotenoids content, photosynthetic rate, enzymatic and nonenzymatic antioxidant activity, and the production of carbohydrates, thus improving the growth and production of plants treated with SA under both normal and environmental stress conditions as compared to untreated plants under stress conditions. Zn is a constituent of various enzymes that facilitate the diffusion of CO_2_ from stomatal apertures to the carboxylation sites by Rubisco and the export of carbohydrates from source to sink [24,62]. It also plays a special role in cell division, nitrogen metabolism, photosynthesis, and the synthesis of protein auxin, RNA, and DNA [63]. The macronutrients present in the combination treatments, particularly NPK, when applied through the foliar spray could have boosted the photosynthesis process and regulated other metabolic processes in the plant [24,64]. K plays a positive role in the accumulation of carbohydrates in source (leaves) and its translocation to sink (grains), thus improving the yield. It is also plays a key role in the maintenance of turgor pressure and osmotic adjustment under stress conditions [14,15,22]. All the abovementioned advantages of treated plants with SA, macronutrients, and micronutrients under both normal and stress conditions might explain why strong and positive correlations between most studied parameters were observed under both the FL and LM regimes.

## 4. Materials and Methods

### 4.1. Experimental Site Describtion and Cultivation Conditions

Two field experiments were conducted from December to April in two successive seasons of 2020/2021 and 2021/2022 at the Agricultural Research Farm of the Plant Production Department, College of Food and Agriculture Sciences, King Saud University, Riyadh, Saudi Arabia (24°25′03″ N 46°39′17″ E, 570 m above sea level) to investigate the effect of the exogenous co-application of SA, macronutrients, and micronutrients through soil and/or foliar spray methods on the morpho-physiological traits, yield attributes, and IWUE of wheat under FL and LM irrigation regimes. The soil texture of the research farm is classified as a sandy loam (57.92% sand, 28.65 % silt, and 13.42% clay), and it is climatically categorized as typical arid climatic conditions. The monthly averages of the meteorological data at the research farm during both growing winter seasons are shown in Table 8.

The soil was ploughed twice, leveled, and divided into plots (4 m × 2 m each). Phosphorus fertilizer was applied during the seedbed preparation at the rate of 90 kg P_2_O_5_ ha^−1^ in the form of calcium superphosphate (17% P_2_O_5_). Then, seeds of the spring wheat cultivar Summit were planted manually in ten rows, each 4 m long and 20 cm apart from the adjacent row, at a seeding rate of 150 kg ha^−1^. The seeds were planted on December 8th and 1st in the first and second seasons, respectively. Other nutrients, potassium (K) and nitrogen (N), were applied at the rate of 60 kg K_2_O and 180 kg N ha^−1^ in the form of potassium sulfate (50% K_2_O) and urea (46% N), respectively. Potassium and nitrogen fertilizers were applied at two and three equal doses, respectively. The first and second doses of both fertilizers were applied at the seedling and stem elongation stages, while the third dose of N was applied at booting stage. Other agronomic practices, such as the control of weeds and disease, were done in a timely manner.

### 4.2. Experimental Design and Treatments

The experiments were accomplished in a randomized complete block design with a split plot arrangement and replicated three times. The two irrigation regimes (FL and LM regimes) were assigned to the main plot, while the different combination treatments were randomly distributed in subplots. The first irrigation regime represents non-water-stressed plants (100% of the estimated crop evapotranspiration; ETc), while the second one represents water-stressed plants (50% ETc). The different combination treatments included a possible combination between the application methods for the co-application of SA, macronutrients (NPK), and micronutrients (Zn and Mn). These treatments included control (T1), the soil application of SA (T2), soil application of SA and micronutrients (T3), soil and foliar application of SA and micronutrients (T4), soil application of SA and foliar application of macronutrients and micronutrients (T5), foliar application of SA (T6), and foliar application of SA, macronutrients, and micronutrients (T7).

Based on the reference evapotranspiration rate (ETo) and crop coefficient (Kc), the amount of irrigation water required (ETc) for the FL regime was calculated using the following Equation:ETc = ETo × Kc(1)

The ETo was determined based on the modified Penman–Monteith equation [65], while the Kc values of spring wheat provided in the FAO-56 were used after adjustment based on the climatic conditions of the research farm. Based on this calculation, the cumulative irrigation volume for the FL regime was 6470 and 6500 m^3^ ha^−1^ for the first and second seasons, respectively. The LM regime received 50% of the irrigation volume of the FL regime. The irrigation water was applied using the modified surface irrigation method, as outlined by El-Hendawy et al. [66]. The irrigation treatments were applied 20 days after sowing. 

The soil application of SA and micronutrients were applied 30 days after sowing. The SA, Zn, and Mn were applied at a rate of 3, 20, and 15 kg ha^−1^, respectively. SA, Zn and Mn were applied in the form of HOC_6_H_4_COOH, ZnSO_4_.7H_2_O, and MnSO_4_.3H_2_O, respectively. The foliar application treatments were applied twice at 40 and 60 days after sowing at the concentrations of 0.5% for Mn, 1% for N, P, K, and Zn, and 2.0 mM for SA. Salicylic acid was dissolved in absolute ethanol, while the other compounds were dissolved in distilled water to make a stock. Then, the appropriate quantities of this stock were gently added to the distilled water to make spray solutions with the required concentrations. The various combinations of SA solution and nutrient elements containing 0.1% Tween-20 were sprayed directly onto the leaves of the plants to the point of runoff using a back-mounted pressurized sprayer (16 L) with a T-jet nozzle that was calibrated to deliver 15 mL s-1 at a pressure of 207 kPa. 

### 4.3. Data Recorded

#### 4.3.1. Growth Parameters

Ten plants were randomly harvested from each plot 100 days after sowing to record SFW, PH, TN, and GLN. Then, the surface area of all green leaf blades was measured using an area meter (LI 3100; LI-COR Inc., Lincoln, NE, USA) to record the leaf area per plant, which was used to calculate the leaf area index (LAI) according to Equation (2). All parts of the ten plants were oven-dried at 80 °C to a constant weight and then weighed to obtain the SDW.
(2)$$LAI=\frac{Leaf\;area\;per\;plant\;({cm}^{2})}{Ground\;area\;({cm}^{2})}$$

#### 4.3.2. Physiological Parameters 

A second leaf of five randomly selected plants was taken to record the RWC using the following equation [67]:(3)$$RWC\;(\%)=\frac{FW-DW}{TW-DW}\times 100$$
where FW is the fresh weight of an area of 7–10 cm^2^ from each leaf, TW is the turgid weight of leaf samples after being rehydrated in distilled water for 24 h in the darkroom at 25 °C, and DW is the dry weight of leaf samples after being dried at 80 °C for 72 h.

The chlorophyll pigments, namely Chl a, Chl b, and Chlt, were determined following the methods of Arnon [68] and Lichtenthaler and Wellburn [69] and calculated using the following Equations: Chl a mg g^−1^ = [(12.7 × A663) − (2.69 × A645)] × V/1000 × FW(4)
Chl b mg g^−1^ = [(22.9 × A645) − (4.68 × A663)] × V/1000 × FW(5)
Chl t mg g^−1^ = [(20.2 × A645) + (8.02 × A663)] × V/1000 × FW(6)
where A is the absorbance at specific wavelengths, V is the final volume of extract (ml), and FW is the fresh weight of tissue extracted (g).

#### 4.3.3. Yield and Yield Components

After the plants reached maturity on 14th April in both seasons, fifty randomly selected spikes from each plot were collected to determine SL, GWS, NGPS, and TGW. To determine BY and GY, five inner rows of 3 m (3.0 m^2^) from each plot were harvested manually, sun-dried, and then weighed to determine BY in kg per 3.0 m^2^ before being converted to ton ha^−1^. All spikes of the harvested area were threshed and the grains were collected, cleaned, dried, and weighed to determine GY in kg per 3.0 m^2^ and then converted to ton ha^−1^. The HI, which is the ratio of GY and BY, was calculated using Equation (7), while the IWUE, which is the ratio of GY and ETc, was calculated using Equation (8) [70]: (7)$$HI\;(\%)=\frac{GY}{BY}\times 100$$
(8)$$IWUE\;(kg\;ha\;{mm}^{-1})=\frac{GY}{ETc}$$

### 4.4. Data Analysis

An analysis of variance (ANOVA) was performed according to the split plot in a randomized complete block design by using the SAS statistical software 9.3 for Windows (SAS Institute, Cary, NC, USA). The mean values of treatments were compared using Duncan’s test at a *p* ≤ 0.05 significance level. For a better understanding of the relationship between the traits across studied factors, Pearson’s correlation coefficient and principle components analysis (PCA) were applied and performed using a computer software program XLSTAT statistical package (vers. 2019.1, Excel add-ins soft SARL, New York, NY, USA). A heatmap was performed using heatmap packages in R statistical software (version 4.2.2). The Pearson’s correlation, PCA, and heatmap were made based on the pooled data of the two seasons. All figures were plotted using SigmaPlot 14 software program (v. 14.0; SPSS, Chicago, IL, USA).

## 5. Conclusions

The results of this study confirmed that it is difficult to grow wheat under deficit irrigation in arid climatic conditions without a significant decrease in its growth and production. Although the LM regime improved the IWUE, it markedly reduced the different morpho-physiological traits, which ultimately lead to a wheat yield and yield components reduction as compared to the FL regime. Nevertheless, the co-application of SA, macro nutrients, and micronutrients through soil and foliar spray methods effectively mitigated the negative impacts of the LM regime, mainly by improving plant growth and physiological attributes. It also enhanced the performance of wheat under the FL regime. Therefore, strong and positive correlations among most studied parameters were observed under both irrigation regimes. In conclusion, combinations of osmolytes compounds, macronutrients, and micronutrients that are exogenously applied through foliar and soil methods could be recommended as an effective strategy to augment the growth and production of wheat crops under both full and deficit irrigation regimes in arid agro-ecosystems. 

## Figures and Tables

**Figure 1 plants-12-01368-f001:**
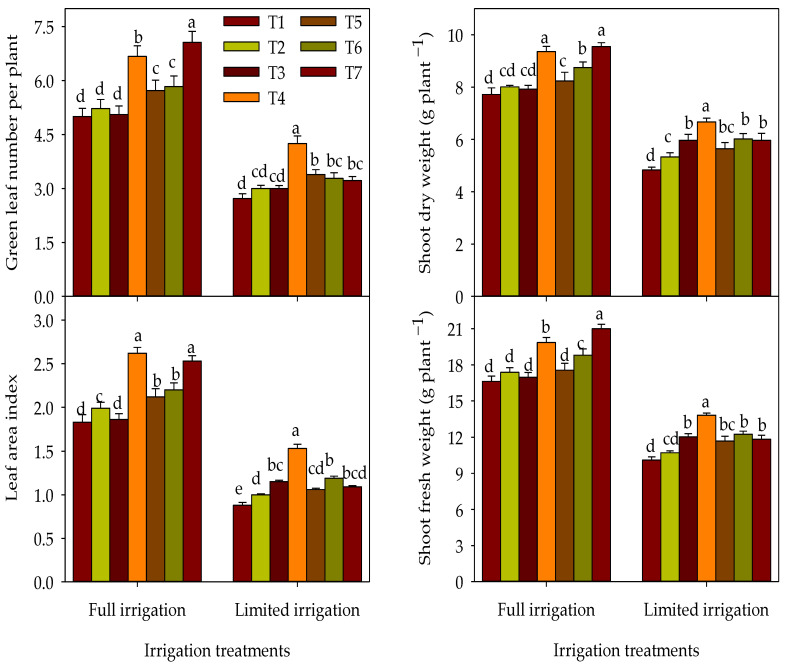
Response of different growth parameters to the interaction between irrigation regimes and treatments over the two growing seasons. The full name of the abbreviations of different combination treatments (T1–T7) is listed in the footer of Table 2. The different letters indicate statistical significance at the 0.05 level.

**Figure 2 plants-12-01368-f002:**
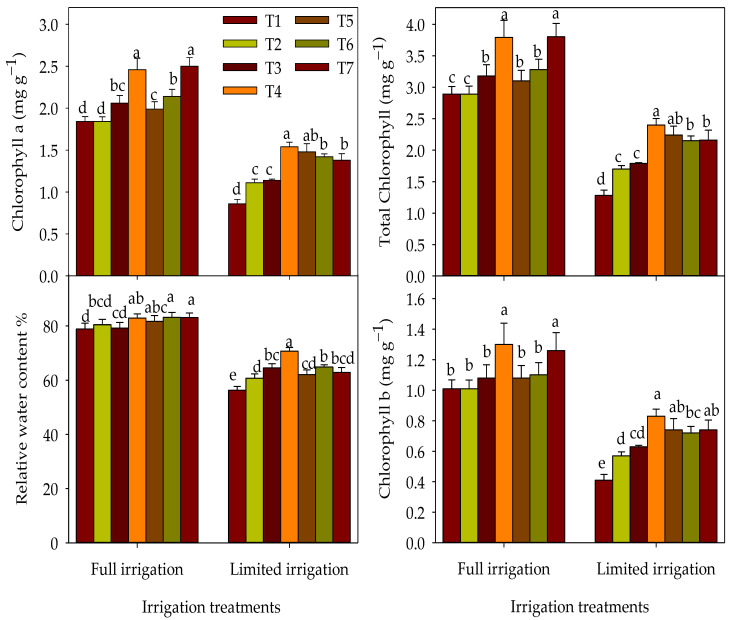
Response of the physiological parameters of wheat to the interaction between irrigation regimes and combination treatments over two growing seasons. The full name of the abbreviations of different combination treatments (T1–T7) is listed in the footer of Table 2. The different letters indicate statistical significance at the 0.05 level, according to Duncan’s test.

**Figure 3 plants-12-01368-f003:**
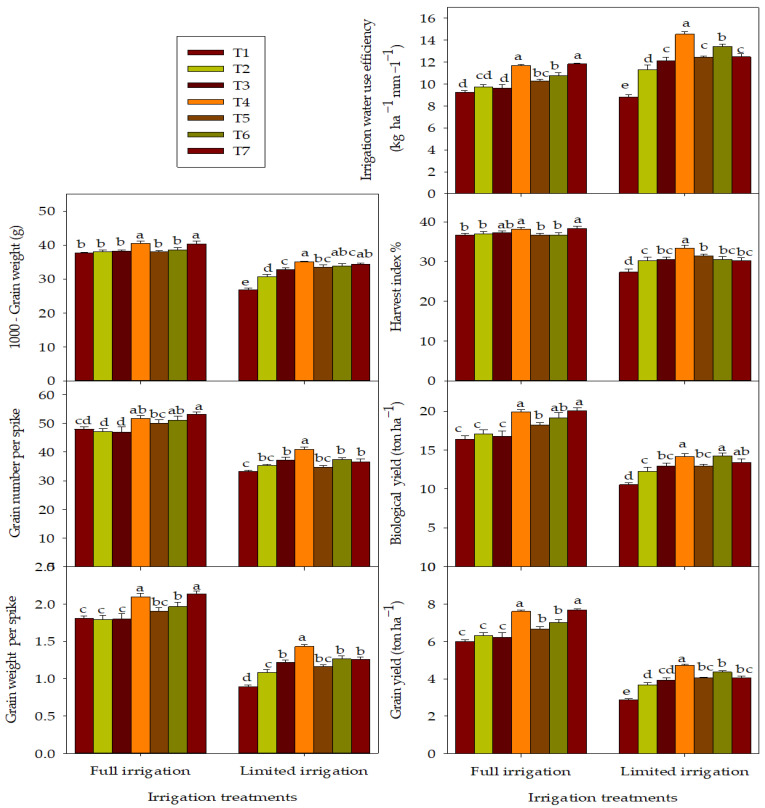
Response of grain yield, yield traits, and irrigation water use efficiency of wheat to the interaction between the irrigation regimes and combination treatments over two growing seasons. The full name of the abbreviations of different combination treatments (T1–T7) is listed in the footer of Table 2. The different letters indicate statistical significance at the 0.05 level.

**Figure 4 plants-12-01368-f004:**
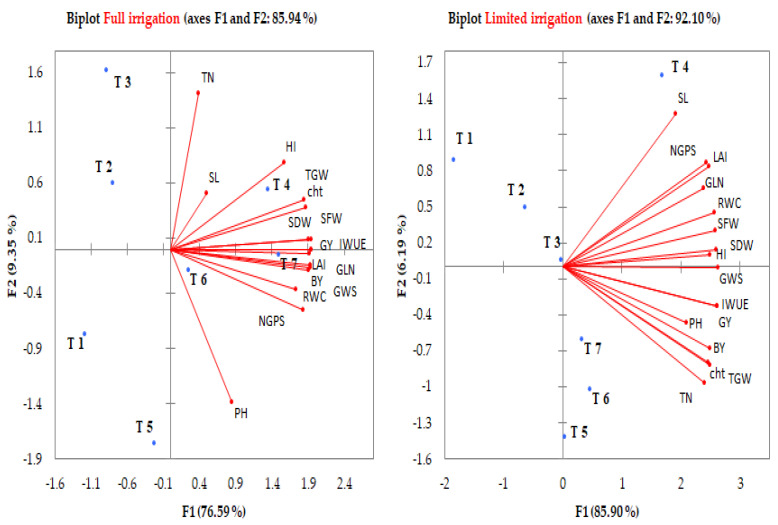
Biplot of the principal component analysis for the first two components of different parameters of wheat and different combination treatments under each irrigation regime. The full name of the abbreviations of different growth, physiological, and yield and irrigation water use efficiency parameters is listed in the footers of Table 1, Table 3 and Table 5, respectively. The full name of the abbreviations of irrigation regimes and different combination treatments is listed in the footer of Table 2.

**Figure 5 plants-12-01368-f005:**
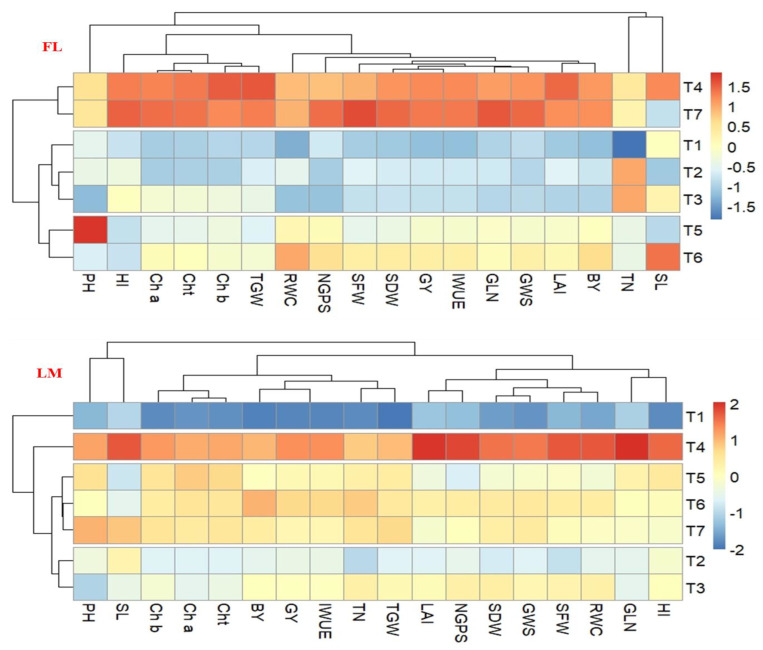
Heatmap of different parameters of wheat and different combination treatments under each irrigation regime. The full name of the abbreviations of different growth, physiological, and yield and irrigation water use efficiency parameters is listed in the footers of Table 1, Table 3 and Table 5, respectively. The full name of the abbreviations of irrigation regimes and different combination treatments is listed in the footer of Table 2.

**Table 1 plants-12-01368-t001:** Analysis of variance (F-values) for the effects of irrigation regimes (IR), combination treatments (T), and their interactions on different plant growth parameters for each season and combined across the two seasons.

**Sources**	**df**	**PH**	**TN**	**GLN**	**LAI**	**SFW**	**SDW**
		First season					
IR	1	950.50 **	2231.40 **	4550.53 **	1878.46 **	3272.48 **	1157.69 **
T	6	3.46 *	2.50 ^ns^	16.17 **	59.30 **	14.26 **	23.44 **
IR × T	6	1.25 ^ns^	1.05 ^ns^	3.88 **	13.08 **	4.05 **	3.35 *
		Second season					
IR	1	76.47 *	20.48 *	210.28 **	942.13 **	507.28 **	284.07 **
T	6	0.65 ^ns^	1.54 ^ns^	42.37 **	41.69 **	14.51 **	17.39 **
IR × T	6	0.43 ^ns^	0.44 ^ns^	14.34 **	9.53 **	3.90 **	3.2 *
		Combined two seasons					
Season (S)	1	4.24 ^ns^	9.86 ^ns^	17.55 ^ns^	16.16 ^ns^	0.01 ^ns^	225.89 **
IR	1	296.25 **	185.57 **	1196.65 **	2688.63 **	1515.43 **	846.36 **
IR × S	1	0.20 ^ns^	22.51 **	49.05 **	40.73 **	15.70 *	3.18 ^ns^
T	6	1.29 ^ns^	3.10 *	42.90 **	96.69 **	27.85 **	39.44 **
T × S	6	1.04 ^ns^	0.60 ^ns^	0.83 ^ns^	0.26 ^ns^	0.90 ^ns^	0.61 ^ns^
T × IR	6	0.38 ^ns^	1.30 ^ns^	10.87 **	19.52 **	7.38 **	6.06 **
T × IR × S	6	0.78 ^ns^	0.33 ^ns^	1.40 ^ns^	2.27 ^ns^	0.58 ^ns^	0.50 ^ns^

Abbreviations in the table indicate plant height (PH), tiller number per plant (TN), green leaf number per plant (GLN), leaf area index (LAI), shoot fresh weight (SFW), and shoot dry weight (SDW). ^ns^ indicates non-significance while * and ** indicate significance at *p* ≤ 0.05 and 0.01, respectively, in F-tests.

**Table 2 plants-12-01368-t002:** Effects of irrigation regimes and combination treatments (T) on growth parameters of wheat in first season (S1), second season (S2), and across both seasons (comb).

T	**S1**	**S2**	**Comb.**	**S1**	**S2**	**Comb.**	**S1**	**S2**	**Comb.**
	PH (cm plant^−1^)			TN			GLN		
T1	74.56 ^c^	72.83 ^ns^	73.69 ^ns^	3.72 ^ns^	3.33 ^ns^	3.53 ^b^	4.17 ^e^	3.56 ^d^	3.86 ^d^
T2	74.42 ^c^	74.33 ^ns^	74.38 ^ns^	4.28 ^ns^	3.72 ^ns^	4.00 ^a^	4.34 ^de^	3.89 ^c^	4.11 ^d^
T3	74.02 ^c^	73.33 ^ns^	73.67 ^ns^	4.39 ^ns^	3.89 ^ns^	4.14 ^a^	4.28 ^e^	3.78 ^cd^	4.03 ^d^
T4	77.17 ^a^	74.06 ^ns^	75.61 ^ns^	4.50 ^ns^	3.72 ^ns^	4.11 ^a^	5.92 ^a^	5.00 ^a^	5.46 ^a^
T5	76.72 ^ab^	74.55 ^ns^	75.64 ^ns^	4.06 ^ns^	3.83 ^ns^	3.95 ^a^	4.78 ^cd^	4.33 ^b^	4.55 ^c^
T6	74.86 ^bc^	74.22 ^ns^	74.54 ^ns^	4.33 ^ns^	3.67 ^ns^	4.00 ^a^	4.84 ^c^	4.28 ^b^	4.56 ^c^
T7	74.67 ^c^	76.39 ^ns^	75.53 ^ns^	4.33 ^ns^	3.78 ^ns^	4.06 ^a^	5.39 ^b^	4.89 ^a^	5.14 ^b^
FL	80.16 ^a^	78.95 ^a^	79.55 ^a^	4.87 ^a^	4.02 ^a^	4.45 ^a^	6.33 ^a^	5.25 ^a^	5.79 ^a^
LI	70.24 ^b^	69.54 ^b^	69.89 ^b^	3.59 ^b^	3.40 ^b^	3.49 ^b^	3.29 ^b^	3.24 ^b^	3.27 ^b^
	LAI			SFW (g plant^−1^)			SDW (g plant^−1^)		
T1	1.44 ^f^	1.27 ^e^	1.35 ^f^	13.34 ^d^	13.38 ^e^	13.36 ^d^	6.55 ^e^	6.00 ^d^	6.27 ^e^
T2	1.56 ^e^	1.43 ^d^	1.49 ^e^	13.85 ^d^	14.24 ^de^	14.04 ^c^	6.87 ^de^	6.46 ^c^	6.67 ^d^
T3	1.57 ^e^	1.44 ^d^	1.50 ^e^	14.24 ^cd^	14.74 ^cd^	14.49 ^c^	7.23 ^cd^	6.66 ^c^	6.95 ^c^
T4	2.14 ^a^	2.01 ^a^	2.07 ^a^	16.70 ^a^	16.97 ^a^	16.83 ^a^	8.26 ^a^	7.76 ^a^	8.01 ^a^
T5	1.68 ^d^	1.50 ^d^	1.59 ^d^	15.00 ^bc^	14.22 ^de^	14.61 ^c^	7.37 ^bc^	6.50 ^c^	6.93 ^cd^
T6	1.77 ^c^	1.62 ^c^	1.69 ^c^	15.43 ^b^	15.62 ^bc^	15.52 ^b^	7.64 ^b^	7.13 ^b^	7.38 ^b^
T7	1.86 ^b^	1.76 ^b^	1.81 ^b^	16.60 ^a^	16.22 ^ab^	16.41 ^a^	8.12 ^a^	7.41 ^ab^	7.76 ^a^
FL	2.30 ^a^	2.03 ^a^	2.16 ^a^	17.96 ^a^	18.65 ^a^	18.30 ^a^	8.71 ^a^	8.30 ^a^	8.51 ^a^
LI	1.13 ^b^	1.12 ^b^	1.13 ^b^	12.09 ^b^	11.46 ^b^	11.77 ^b^	6.15 ^b^	5.40 ^b^	5.77 ^b^

The full name of the abbreviations of different growth parameters is listed in the footer of Table 1. T1, control; T2, soil application of salicylic acid; T3, soil application of salicylic acid and micronutrients; T4, soil and foliar application of salicylic acid and micronutrients; T5, soil application of salicylic acid and foliar application of macronutrients and micronutrients; T6, foliar application of salicylic acid; T7, foliar application of salicylic acid, macronutrients, and micronutrients; FL, full irrigation regime; LM, limited irrigation regime (LM). Values followed by the same letter in the same column are not significantly different at 0.05 level according to Duncan’s test. ^ns^, denotes non-significance.

**Table 3 plants-12-01368-t003:** Analysis of variance (F-values) for the effects of irrigation regimes (IR), combination treatments (T), and their interaction on different plant physiological parameters for each season and combined across the two seasons.

**Sources**	**df**	**RWC**	**Ch a**	**Ch b**	**Ch t**
		First season			
IR	1	1608.41 **	5150.56 **	265.56 **	1848.98 **
T	6	9.65 **	16.93 **	4.25 **	11.45 **
IR × T	6	2.61 *	3.96 **	0.92 ^ns^	2.38 *
		Second season			
IR	1	687.35 **	509.47 **	365.75 **	464.09 **
T	6	11.93 **	60.55 **	21.32 **	62.50 **
IR × T	6	6.86 **	11.63 **	2.74 *	10.02 **
		Combined two seasons			
Season (S)	1	614.80 **	177.96 **	65.91 *	486.06 **
IR	1	1797.19 **	1436.15 **	619.84 **	1147.65 **
IR × S	1	11.87 *	37.54 **	53.22 **	49.85 **
T	6	20.02 **	68.60 **	24.41 **	60.92 **
T × S	6	0.83 ^ns^	6.40 **	7.27 **	8.56 **
T × IR	6	7.60 **	12.53 **	3.09 *	9.15 **
T × IR × S	6	0.49 ^ns^	2.62 *	1.22 ^ns^	2.58 ^ns^

Abbreviations in the table indicate the relative water content (RWC), chlorophyll a (Cha), chlorophyll b (Chb), and total chlorophyll content (Cht). ^ns^ indicates non-significance while * and ** indicate significance at *p* ≤ 0.05 and 0.01, respectively, in F-tests.

**Table 4 plants-12-01368-t004:** Effects of irrigation regimes and combination treatments (T) on physiological parameters of wheat in the first season (S1), second season (S2), and across both seasons (comb).

T	S1	S2	Comb	S1	S2	Comb	S1	S2	Comb	S1	S2	Comb
	RWC (%)			Cha (mg g^−1^ FW)			Chlb (mg g^−1^ FW)			Cht (mg g^−1^ FW)		
T1	63.85 ^e^	71.29 ^d^	67.57 ^e^	1.34 ^e^	1.36 ^e^	1.35 ^e^	0.69 ^c^	0.73 ^c^	0.71 ^d^	2.05 ^e^	2.13 ^e^	2.09 ^e^
T2	67.24 ^d^	73.94 ^c^	70.59 ^d^	1.44 ^de^	1.51 ^d^	1.48 ^de^	0.75 ^bc^	0.83 ^c^	0.79 ^cd^	2.21 ^de^	2.38 ^d^	2.30 ^de^
T3	68.20 ^cd^	75.44 ^bc^	71.82 ^cd^	1.52 ^d^	1.69 ^c^	1.60 ^cd^	0.76 ^bc^	0.96 ^b^	0.86 ^bcd^	2.30 ^cd^	2.66 ^c^	2.48 ^cd^
T4	73.75 ^a^	79.78 ^a^	76.76 ^a^	1.81 ^a^	2.19 ^a^	2.00 ^a^	0.87 ^a^	1.26 ^a^	1.06 ^a^	2.71 ^a^	3.48 ^a^	3.10 ^a^
T5	67.83 ^cd^	75.81 ^bc^	71.82 ^cd^	1.56 ^cd^	1.91 ^b^	1.74 ^bc^	0.78 ^b^	1.04 ^b^	0.91 ^bc^	2.36 ^cd^	2.98 ^b^	2.67 ^c^
T6	71.57 ^ab^	76.42 ^b^	74.00 ^b^	1.66 ^bc^	1.90 ^b^	1.78 ^b^	0.78 ^b^	1.03 ^b^	0.90 ^bc^	2.47 ^bc^	2.96 ^b^	2.71 ^bc^
T7	70.34 ^bc^	75.59 ^bc^	72.96 ^bc^	1.77 ^ab^	2.11 ^a^	1.94 ^a^	0.81 ^ab^	1.19 ^a^	1.00 ^ab^	2.61 ^ab^	3.34 ^a^	2.98 ^ab^
FL	77.32 ^a^	85.30 ^a^	81.31 ^a^	1.94 ^a^	2.30 ^a^	2.12 ^a^	0.94 ^a^	1.30 ^a^	1.12 ^a^	2.90 ^a^	3.64 ^a^	3.27 ^a^
LI	60.62 ^b^	65.64 ^b^	63.13 ^b^	1.23 ^b^	1.32 ^b^	1.28 ^b^	0.61 ^b^	0.71 ^b^	0.66 ^b^	1.87 ^b^	2.05 ^b^	1.96 ^b^

The full name of the abbreviations of different physiological parameters is listed in the footer of Table 3. The full name of the abbreviations of the irrigation regimes and different combination treatments is listed in the footer of Table 2. The different letters indicate statistical significance at the 0.05 level, according to Duncan’s test.

**Table 5 plants-12-01368-t005:** Analysis of variance (F-values) for the effects of irrigation regimes (IR), combination treatments (T), and their interaction on yield components, yield, and irrigation water use efficiency for each season and combined across the two seasons.

**Sources**	**df**	**SL**	**GWS**	**NGPS**	**TGW**	**GY**	**BY**	**HI**	**IWUE**
		First season							
IR	1	941.71 **	990.33 **	695.44 **	745.57 **	4143.05 **	300.84 **	228.37 **	289.67 **
T	6	0.66 ^ns^	19.95 **	8.80 **	21.45 **	30.51 **	14.48 **	12.03 **	34.19 **
IR × T	6	0.30 ^ns^	4.82 **	4.20 **	9.15 **	4.17 **	2.00 ^ns^	7.36 **	6.52 **
		Second season							
IR	1	210.99 **	1743.81 **	882.81 **	348.83 **	866.24 **	905.52 **	1303.84 **	136.20 **
T	6	0.35 ^ns^	10.34 **	3.07 *	7.58 **	18.61 **	15.65 **	6.46 **	27.85 **
IR × T	6	0.67 ^ns^	1.20 ^ns^	1.03 ^ns^	2.54 *	2.72 *	2.41 *	2.39 *	6.27 **
		Combined two seasons							
Season (S)	1	0.19 ^ns^	718.11 **	80.67*	15.82 ^ns^	1.20 ^ns^	15.16 ^ns^	664.43 **	1.90 ^ns^
IR	1	588.94 **	2574.20 **	1556.89 **	912.13 **	2875.86 **	905.91 **	765.10 **	355.59 **
IR × S	1	8.88 ^ns^	3.33 ^ns^	0.01 ^ns^	2.67 ^ns^	0.02 ^ns^	0.01 ^ns^	0.10 ^ns^	1.05 ^ns^
T	6	0.67 ^ns^	26.84 **	9.01 **	22.14 **	47.00 **	29.14 **	15.82 **	61.28 **
T × S	6	0.20 ^ns^	0.39 ^ns^	0.46 ^ns^	0.97 ^ns^	0.57 ^ns^	1.04 ^ns^	4.02 **	0.75 ^ns^
T × IR	6	0.54 ^ns^	4.24 **	3.25 **	7.58 **	6.63 **	3.28 **	7.84 **	12.32 **
T × IR × S	6	0.48 ^ns^	0.55 ^ns^	0.66 ^ns^	1.28 ^ns^	0.34 ^ns^	1.14 ^ns^	3.10*	0.47 ^ns^

Abbreviations in the table indicate spike length (SL), grain weight per spike (GWS), number of grains per spike (NGPS), thousand grain weight (TGW), grain yield per ha (GY), biological yield per ha (BY), harvest index (HI), and irrigation water use efficiency (IWUE). ^ns^ indicates non-significance while * and ** indicate significance at *p* ≤ 0.05 and 0.01, respectively, in F-tests.

**Table 6 plants-12-01368-t006:** The effects of irrigation regimes and combination treatments (T) on yield components, yield, and irrigation water use efficiency of wheat in first season (S1), second season (S2), and across both seasons (Comb).

**T**	**S1**	**S2**	**Comb**	**S1**	**S2**	**Comb**	**S1**	**S2**	**Comb**	**S1**	**S2**	**Comb**
	SL (cm)			GWS (g)			GNPS			TGW (g)		
T1	8.17 ^ns^	8.27 ^ns^	8.22 ^ns^	1.32 ^f^	1.39 ^e^	1.35 ^e^	40.0 ^d^	41.3 ^d^	40.65 ^d^	31.9 ^e^	32.60 ^c^	32.26 ^d^
T2	8.29 ^ns^	8.24 ^ns^	8.27 ^ns^	1.37 ^ef^	1.51 ^de^	1.44 ^d^	40.3 ^d^	42.1 ^cd^	41.20 ^d^	33.3 ^d^	35.27 ^b^	34.29 ^c^
T3	8.29 ^ns^	8.27 ^ns^	8.28 ^ns^	1.43 ^de^	1.58 ^cd^	1.51 ^cd^	40.0 ^d^	44.1 ^abc^	42.07 ^d^	35.5 ^c^	35.49 ^b^	35.49 ^b^
T4	8.62 ^ns^	8.46 ^ns^	8.54 ^ns^	1.72 ^a^	1.81 ^a^	1.76 ^a^	45.7 ^a^	46.95 ^a^	46.32 ^a^	37.3 ^a^	38.23 ^a^	37.77 ^a^
T5	8.27 ^ns^	8.09 ^ns^	8.18 ^ns^	1.50 ^cd^	1.56 ^cd^	1.53 ^c^	41.2 ^cd^	43.5 ^bcd^	42.34 ^cd^	36.0 ^bc^	35.48 ^b^	35.74 ^b^
T6	8.38 ^ns^	8.32 ^ns^	8.35 ^ns^	1.55 ^bc^	1.68 ^bc^	1.61 ^b^	43.1 ^bc^	45.4 ^abc^	44.24 ^bc^	35.7 ^c^	36.5 ^ab^	36.12 ^b^
T7	8.27 ^ns^	8.41 ^ns^	8.34 ^ns^	1.64 ^ab^	1.75 ^ab^	1.69 ^a^	43.8 ^ab^	45.8 ^ab^	44.8 ^ab^	37.0 ^ab^	37.60 ^a^	37.31 ^a^
FL	8.97 ^a^	9.12 ^a^	9.04 ^a^	1.86 ^a^	1.99 ^a^	1.93 ^a^	48.59 ^a^	50.76 ^a^	49.67 ^a^	38.3 ^a^	39.25 ^a^	38.76 ^a^
LI	7.68 ^b^	7.47 ^b^	7.59 ^b^	1.15 ^b^	1.23 ^b^	1.19 ^b^	35.44 ^b^	37.55 ^b^	36.50 ^b^	32.2 ^b^	32.52 ^b^	32.38 ^b^
	GY (ton ha^−1^)			BY (ton ha^−1^)			HI (%)			IWUE (kg ha^−1^ mm^−1^)		
T1	4.33 ^e^	4.54 ^e^	4.44 ^e^	13.24 ^c^	13.68 ^c^	13.46 ^d^	31.68 ^c^	32.29 ^b^	31.99 ^c^	8.83 ^e^	9.24 ^d^	9.03 ^e^
T2	4.89 ^d^	5.11 ^d^	5.00 ^d^	13.77 ^c^	15.57 ^b^	14.67 ^c^	34.88 ^b^	32.31 ^b^	33.59 ^b^	10.23 ^d^	10.82 ^c^	10.52 ^d^
T3	4.97 ^d^	5.19 ^d^	5.08 ^d^	14.07 ^c^	15.61 ^b^	14.84 ^c^	34.91 ^b^	32.83 ^b^	33.87 ^b^	10.63 ^d^	11.07 ^c^	10.85 ^d^
T4	6.12 ^a^	6.21 ^a^	6.17 ^a^	16.50 ^a^	17.64 ^a^	17.07 ^a^	36.72 ^a^	34.81 ^a^	35.76 ^a^	13.00 ^a^	13.24 ^a^	13.12 ^a^
T5	5.40 ^c^	5.31 ^cd^	5.36 ^c^	15.25 ^b^	15.88 ^b^	15.57 ^b^	34.95 ^b^	33.02 ^b^	33.99 ^b^	11.41 ^c^	11.29 ^c^	11.35 ^c^
T6	5.71 ^bc^	5.65 ^bc^	5.68 ^b^	16.23 ^ab^	17.21 ^a^	16.72 ^a^	34.84 ^b^	32.40 ^b^	33.62 ^b^	12.18 ^b^	11.99 ^b^	12.09 ^b^
T7	5.82 ^ab^	5.91 ^ab^	5.87 ^b^	15.98 ^ab^	17.53 ^a^	17.75 ^a^	35.58 ^ab^	32.98 ^b^	34.28 ^b^	12.0 ^bc^	12.27 ^b^	12.15 ^b^
FL	6.74 ^a^	6.83 ^a^	6.79 ^a^	17.65 ^a^	18.82 ^a^	18.24 ^a^	38.19 ^a^	36.27 ^a^	37.23 ^a^	10.37 ^b^	10.51 ^b^	10.44 ^b^
LI	3.90 ^b^	4.00 ^b^	3.95 ^b^	12.36 ^b^	13.50 ^b^	12.93 ^b^	31.39 ^b^	29.63 ^b^	30.51 ^b^	11.99 ^a^	12.32 ^a^	12.16 ^a^

The full name of the abbreviations of different yield parameters is listed in the footer of Table 5. The full name of the abbreviations of irrigation regimes and different combination treatments is listed in the footer of Table 2. The different letters indicate statistical significance at the 0.05 level according to Duncan’s test. ^ns^ denotes non-significance.

**Table 7 plants-12-01368-t007:** Correlation matrix for different parameters (P) of growth, physiological, yield, and IWUE of wheat under full irrigation (upper right) and limited irrigation (lower left) regimes over both growing seasons.

P	1	2	3	4	5	6	7	8	9	10	11	12	13	14	15	16
PH (1)		−0.16	0.50	0.49	0.33	0.36	0.47	0.29	−0.34	0.45	0.52	0.29	0.44	0.47	0.16	0.44
TN (2)	0.70		0.15	0.21	0.21	0.21	0.14	0.27	−0.13	0.07	−0.10	0.31	0.22	0.16	0.43	0.22
GLN (3)	**0.79**	0.68		**0.97**	**0.98**	**0.98**	**0.86**	**0.93**	0.13	**0.99**	**0.95**	**0.93**	**0.98**	**0.96**	**0.81**	**0.98**
LAI (4)	0.62	0.74	**0.93**		**0.95**	**0.97**	**0.89**	**0.92**	0.25	**0.97**	**0.91**	**0.94**	**0.99**	**0.97**	**0.79**	**0.99**
SFW (5)	0.67	**0.87**	**0.90**	**0.97**		**0.99**	**0.87**	**0.93**	0.18	**0.97**	**0.93**	**0.92**	**0.98**	**0.95**	**0.81**	**0.98**
SDW (6)	0.69	**0.91**	**0.84**	**0.94**	**0.98**		**0.89**	**0.95**	0.27	**0.99**	**0.94**	**0.94**	**0.99**	**0.97**	**0.80**	**0.99**
RWC (7)	0.63	**0.84**	**0.88**	**0.98**	**0.98**	**0.98**		**0.75**	0.29	**0.87**	**0.91**	0.72	**0.91**	**0.96**	0.47	**0.91**
Cht (8)	**0.90**	**0.91**	**0.83**	**0.76**	**0.85**	**0.85**	**0.81**		0.33	**0.95**	**0.85**	**0.98**	**0.94**	**0.89**	**0.89**	**0.94**
SL (9)	0.70	0.44	0.74	0.74	0.67	0.70	0.71	0.56		0.27	0.25	0.28	0.26	0.30	0.06	0.26
GWS (10)	0.74	**0.92**	**0.85**	**0.92**	**0.97**	**0.99**	**0.97**	**0.89**	0.72		**0.97**	**0.93**	**0.98**	**0.97**	**0.77**	**0.98**
NGPS (11)	0.55	0.75	**0.84**	**0.98**	**0.95**	**0.95**	**0.98**	0.70	**0.77**	**0.93**		**0.80**	**0.94**	**0.96**	0.60	**0.94**
TGW (12)	**0.81**	**0.97**	0.73	0.75	**0.86**	**0.91**	**0.85**	**0.95**	0.58	**0.94**	**0.75**		**0.93**	**0.87**	**0.93**	**0.93**
GY (13)	**0.77**	**0.93**	**0.85**	**0.88**	**0.94**	**0.95**	**0.94**	**0.94**	0.63	**0.97**	**0.87**	**0.95**		**0.99**	**0.78**	**0.99**
BY (14)	0.72	**0.96**	0.73	**0.80**	**0.88**	**0.92**	**0.88**	**0.91**	0.55	**0.95**	**0.82**	**0.95**	**0.97**		0.67	**0.99**
HI (15)	0.75	**0.79**	**0.91**	**0.88**	**0.90**	**0.89**	**0.92**	**0.89**	0.67	**0.91**	**0.83**	**0.86**	**0.93**	**0.83**		**0.78**
IWUE (16)	**0.77**	**0.93**	**0.85**	**0.88**	**0.94**	**0.95**	**0.94**	**0.94**	0.63	**0.97**	**0.87**	**0.95**	**0.99**	**0.97**	**0.93**	

The full name of the abbreviations of different growth, physiological, and yield and irrigation water use efficiency parameters is listed in the footers of Table 1, Table 3 and Table 5, respectively. Values in bold indicate a significance level alpha = 0.05.

**Table 8 plants-12-01368-t008:** Monthly averages of the climatic data at the research farm during the growing period of wheat in the first (S1) and second (S2) seasons.

Months	Minimum Temperature (°C)		Maximum Temperature (°C)		Relative Humidity (%)		Precipitation (mm)	
	S1	S2	S1	S2	S1	S2	S1	S2
December	9.79	13.23	23.03	23.85	48.19	37.15	2.27	2.08
January	8.29	13.68	22.37	21.52	41.63	45.54	7.31	33.20
February	10.02	15.83	24.15	24.74	40.26	31.59	2.94	0.00
March	15.02	16.99	32.38	30.60	20.10	20.91	0.00	1.17
April	19.95	15.93	35.52	36.55	16.65	18.80	3.44	0.00

## Data Availability

All data are presented within the article.

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
