# Peer review of "Appropriate Application Methods for Salicylic Acid and Plant Nutrients Combinations to Promote Morpho-Physiological Traits, Production, and Water Use Efficiency of Wheat under Normal and Deficit Irrigation in an Arid Climate"

_plants, 2023, doi:10.3390/plants12061368_

Round 1

Reviewer 1 Report

Lines 67 - 100 and 620 - 634 similarly describe the role of macro and micronutrients in the plant, moreover, this is commonly known information

Lines 87, 95 there is no need to repeat many roles, it is enough to describe them

Results - it should be reconsidered whether there is a need to provide all abbreviations of the analyzed parameters under subsequent tables and graphs

Unify citations in referenses - date bolding, journal names

Author Response

Reviewer   #1

We greatly appreciate your critical observations as well as your constructive and helpful comments. We hope that we could address your questions/comments by the explanations and revisions made in the manuscript. We believe that the manuscript is substantially improved after making the suggested revisions.

  • Lines 67 - 100 and 620 - 634 similarly describe the role of macro and micronutrients in the plant, moreover, this is commonly known information

Response: Thank you very much for this remark. The repeated information about macro and micronutrients in the introduction section (lines 620-634) has been shortened.

  • Lines 87, 95 there is no need to repeat many roles, it is enough to describe them

Response: Thank you very much for this remark. The section related to the function of macro- and micro-nutrients in plants in the introduction section has been modified and improved.

  • Results - it should be reconsidered whether there is a need to provide all abbreviations of the analyzed parameters under subsequent tables and graphs

Response: Thank you very much for your comment. The full name of different abbreviations has been written only when mentioned the first time with the first table and figure and not repeated in the sequence Tables and Figures.

  • Unify citations in references - date bolding, journal names

Response: Thank you very much for your comment. The Reference section has been revised according to the guidelines of the Journal.

Reviewer 2 Report

Dear Authors

A two-year field study was undertaken to assess the impact of seven treatments for the integrated application of salicylic acid (SA), macro- and micronutrients on the morpho-physiological traits, yield, and irrigation water use efficiency (IWUE) of wheat subjected to full (FL) and limited (LM) irrigation regimes.

- The introduction must be drastically reduced, it is redundant and excessively generic.

- delete figure 5. it does not report new information to the text

- delete 562-566 lines

- the results must be presented in more scientific and less narrative language. Furthermore, greater synthesis is needed in the periphrases

Author Response

Reviewer   #2

A two-year field study was undertaken to assess the impact of seven treatments for the integrated application of salicylic acid (SA), macro- and micronutrients on the morpho-physiological traits, yield, and irrigation water use efficiency (IWUE) of wheat subjected to full (FL) and limited (LM) irrigation regimes.

 Response: We greatly appreciate your critical observations as well as your constructive and helpful comments. We hope that we could address your questions/comments through the explanations and revisions made in the manuscript. We believe that the manuscript is substantially improved after making the suggested revisions.

  • The introduction must be drastically reduced, it is redundant and excessively generic.

Response: Thank you very much for this remark. The introduction section has been revised, focused, and shortened.

  • Delete figure 5. it does not report new information to the text

Response: Thank you very much for this comment. Figure 5 is very important to show the close relationships between measured parameters and different treatments.

  • delete 562-566 lines

Response: The lines 562-566 in discussion section have been deleted.

  • The results must be presented in more scientific and less narrative language. Furthermore, greater synthesis is needed in the periphrases

Response: Thank you very much for this remark. The result section has been revised accordingly.

Round 2

Reviewer 2 Report

Accepted